# Biodiversity, Ecology, and Secondary Metabolites Production of Endophytic Fungi Associated with Amaryllidaceae Crops

**Gianluca Caruso [1], Nadezhda Golubkina [2], Alessio Tallarita [1], Magdi T. Abdelhamid [3]** 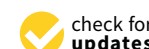 **and Agnieszka Sekara [4,\*]**

[1] Department of Agricultural Sciences, University of Naples Federico II, 80055 Portici (Naples), Italy; gcaruso@unina.it (G.C.); alessio.tallarita@yahoo.com (A.T.)

[2] Federal Scientific Center of Vegetable Production, Selectsionnaya 14 VNIISSOK, 143072 Moscow, Odintsovo, Russia; segolubkina@rambler.ru

[3] Botany Department, National Research Centre, 33 El Behouth Steet, Dokki, Cairo 12622, Egypt; mt.abdelhamid@nrc.sci.eg

[4] Department of Horticulture, Faculty of Biotechnology and Horticulture, University of Agriculture, 31-120 Krakow, Poland

\* Correspondence: agnieszka.sekara@urk.edu.pl; Tel.: +48-12-6625216

**Abstract:** Amaryllidaceae family comprises many crops of high market potential for the food and pharmaceutical industries. Nowadays, the utilization of plants as a source of bioactive compounds requires the plant/endophytic microbiome interactions, which affect all aspects of crop's quantity and quality. This review highlights the taxonomy, ecology, and bioactive chemicals synthesized by endophytic fungi isolated from plants of the Amaryllidaceae family with a focus on the detection of pharmaceutically valuable plant and fungi constituents. The fungal microbiome of Amaryllidaceae is species- and tissue-dependent, although dominating endophytes are ubiquitous and isolated worldwide from taxonomically different hosts. Root sections showed higher colonization as compared to bulbs and leaves through the adaptation of endophytic fungi to particular morphological and physiological conditions of the plant tissues. Fungal endophytes associated with Amaryllidaceae plants are a natural source of ecofriendly bioagents of unique activities, with special regard to those associated with Amaryloidae subfamily. The latter may be exploited as stimuli of alkaloids production in host tissues or can be used as a source of these compounds through in vitro synthesis. Endophytes also showed antagonistic potential against fungal, bacterial, and viral plant diseases and may find an application as alternatives to synthetic pesticides. Although Amaryllidaceae crops are cultivated worldwide and have great economic importance, the knowledge on their endophytic fungal communities and their biochemical potential has been neglected so far.

**Keywords:** onions; amaryllis; endosphere; endobiome; metabolome; symbiosis

## 1. Introduction

Amaryllidaceae species have been utilized as vegetables, herbs, spices, and ornamentals in all continents since ancient times. Many of them have shown widespread benefits in cuisine and healing of common diseases like atherosclerosis, diabetes, inflammation, hypertension, and cancer. These protective effects appear to be related to the presence of organosulfur compounds, predominantly allyl derivatives in Allioidae and alkaloids in the Amaryllidoidae subfamily [1]. The economic significance of Amaryllidaceae crops is substantiated by the all-year-round supply and a wide range of cultivars and landraces characterized by plant parts with different shapes and specific taste and flavor.

The biochemical profile of plants determines a broad or specialized usage by food and pharmacological industry [2–4]. Considering the wide distribution, unique chemical composition, and economic importance, we selected Amaryllidaceae crops to analyze their relationship with the microbiome colonizing the root and shoot tissues.

With regard to root fungal microbiome, Amaryllidaceae symbiosis with arbuscular mycorrhizal fungi has been widely investigated at the physiological, biochemical, and genetic level, and it has been comprehensively reviewed [5–8]. However, there is selective and narrow knowledge of fungal endophytes colonizing these plants. Hardoim et al. [9] defined "endophytes" as microorganisms living in plant tissues in some periods of their life cycle or all through, so the word is referred to the habitat rather than function. Nevertheless, endophytic fungi live inside the tissues of apparently healthy and asymptomatic hosts. The case of Amaryllidaceae crops can help in understanding how fungal endophytes modulate the physiological processes in the above- and below-ground plant tissues as well as interact with a host plant and with other microbes. The effect of plant morphology on colonization and biodiversity of endophyte communities has been rarely studied. The root-associated fungi play an important role in plant nutrition by: mobilizing soil nutrients; recycling organic matter; increasing the water holding capacity and absorption; protecting against pathogens and abiotic stresses; biosynthesizing lignan, auxins, and ethylene, in order to improve the root system expansion and encourage plant fitness through thermotolerance improvement; and coping with salinity, drought, and heavy metal stress [10,11]. The underground bulb has been an additional component of plant tissue/endophyte puzzles as a perennial storage part rich in nutrient compounds, and—in some taxons containing unique bioagents—Amaryllidaceae alkaloids. Taking into account their life cycle, Amaryllidaceae have to efficiently allocate resources from bulbs to vegetative and generative tissues and vice versa during a short vegetation period. It is possible that the assimilates' flow provides a natural path for plants to partitioning the symbiotic endophytes as it was proposed for mycorrhizal fungi by Crişan et al. [12]. Foliar endophytes can stimulate host plant biomass, yield, mineral status, nutritive value, stomatal movement, and defensive mutualism against herbivores and pathogens [13,14].

The strict selection and recruitment of the endophytes by a host species was postulated and analyzed for plant-associated bacteria [15] but the mentioned mechanisms also shape fungal diversity and abundance [16]. The rhizoplane and phylloplane are considered to be a selective gate for endophytes, mediating dynamic changes in the fungal community in plant internal tissues. This is the reason for differences in microbial compositions among host species and even among their tissues [8]. Plant secondary metabolites are postulated as main regulators in colonization and development of fungal endophytes in different tissues and, vice versa, fungal metabolites can play a similar role with respect to the host and the microbiota [17,18]. Plant secondary metabolites are postulated as main regulators in colonization and development of fungal endophytes in different tissues and, vice versa, fungal metabolites can play a similar role with respect to the host and the microbiota [19]. Bioactive metabolites synthesized by fungal endophytes show multidirectional actions in plants and can be explored for the cultivation of targeted functional and medicinal crops [20]. Nowadays, endophytes are also recognized as an alternative to synthetic chemicals in crop production and protection [21]. However, to be successfully applied, endophytes need specific conditions for efficient host tissue colonization. This can be achieved by a thorough understanding of the endophyte-host relationship starting from the colonization up to well-established symbiosis, on the basis of the fungi and plant biology, covering genetic and biochemical factors as well [22–24].

This review aimed to analyze fungal endophyte communities associated with Amaryllidaceae crops, by: presenting the evidence of ubiquitous and/or species- and tissue-dependent fungal microbiome, reviewing the known chemical compounds synthesized by fungal endophytes, and emphasizing their possible vital effects on human proecological and prohealth activities.

## 2. Amaryllidaceae Crops—Botanical Characteristics and Biochemical Composition

According to the current taxonomy, the family Amaryllidaceae consists of three subfamilies, Agapanthoideae, Allioideae, and Amaryllidoideae, comprised of about 80 genera and approximately 2200 accepted species [25]. Amaryllidaceae are perennial or biennial geophytes or hemicryptophytes, with very diversified morphology of their underground shoots, which let distinguish three biomorphological groups: rhizomatous, bulbous, and domesticated onions [26]. In the case of a rhizomatous group (*Cryptostephanus* spp., *Clivia* spp., and some *Scadoxus* spp.), fleshy rhizomes act as storage organs, growing for several years through the successive development of the basal plate. Bulbs, composed of leaf sheaths of varying thicknesses, are formed by rhizomes. The leaves are evergreen. In the bulb group, the true bulb consists of a longitudinally compressed basal stem and fleshy, succulent, storage leaf bases. This group is well-adapted to arid and semiarid climate. Domesticated onions form storage bulbs and were grouped separately because of the diversified morphology, shaped through many centuries of breeding [26]. Most of the Amaryllidaceae plants prefer xerophytic ecosystems, with warm, dry summers and cool, wet winters, being distributed across regions, which are well-recognized biodiversity hotspots. The major center of genetic diversity is localized in Central Asia and Mediterranean basin, and the secondary one, in South Africa, western North America, and the Andes [27]. The gene pool of wild Amaryllidaceae is very rich in the centers of origin and has been explored as a source of new genes introduced into cultivated species and for the domestication of new crops useful as vegetables, herbs, medicinal plants, and ornamentals [28].

Cultivation practices were developed independently in particular regions of the Northern Hemisphere and applied to at least 20 native or introduced vegetables of Allioideae subfamily, especially of genus *Allium*. Onion (*Allium cepa* L.) and garlic are cultivated worldwide, leek (*A. ampeloprasum* L.), shallot (*A. cepa* L. Aggregatum group), and chive (*A. schoenoprasum* L.) dominate in Western and Northern Europe, kurrat (*A. ampeloprasum* L.) in Egypt and the Eastern Mediterranean, Japanese bunching onion (*A. fistulosum* L.) in Japan, rakkyo (*A. chinense* G. Don), and Chinese leek (*A. tuberosum* Rottl. ex Spr.) in China. The cultivated onion group is the result of intensive breeding and represents morphological and physiological characteristics appreciated both for cultivation and marketing. The latter include diversified shape, color, pungency, and chemical composition of bulbs, reduced bolting, long shafts in leek, fast leaf growth in chives, single heart in onion but separated in shallot [29–32]. Onions are known as a major food for preventing chronic disease [33], as a source of sulfur compounds, steroidal saponins, and flavonoids. Moreover, showing a functional food activity, they significantly contribute to the prevention of inflammatory and common lifestyle diseases [34]. Organic sulfur compounds, determining onions' pungent flavor, are the key components responsible for the therapeutic effects [35]: allicin, and ajoene, as well as volatile compounds have the ability to act as antimicrobials and antioxidants [36,37]; sulfur and phenolic compounds also show antioxidant, anticancer, anti-inflammatory activities, and can prevent chronic diseases [38]; quercetin, a bioflavonoid of onions, reveals antiproliferative and proapoptotic effects in many cancer cells, acts as a neuroprotector, and stimulates cellular defense against oxidative stress [39].

*Agapanthus* is the only genus in the subfamily Agapanthoideae, endemic in South Africa but naturalized around the world as ornamental. The Amaryllidoideae have a pronounced floricultural importance because this subfamily comprises popular ornamentals, including many spring-flowering bulbs (*Narcisuss* spp., *Galanthus* spp., and *Leucojum* spp.). They have been grown in European gardens since the ancient times, supplemented since 17–18 century with species of New World origin like *Hippeastrum* spp., or South African, like *Amaryllis* spp. or *Clivia* spp., widely cultivated as indoor plants. The Amaryllidoideae have been traditionally used as medicines to treat mental problems, primarily in Southern Africa [40]. Amaryllidoideae are the source of the isoquinoline alkaloids of unique structure, which were isolated from about 350 species, amongst more than 800 species belonging to this subfamily. Approximately 600 structurally diverse alkaloids were isolated to the date, chemically defined, and pharmacologically investigated, as possessing antibacterial, antifungal, antimalarial, antiviral, antitumor, analgesic, and acetylcholinesterase inhibitory activities [41–43].

The galanthamine was approved to date as the main treatment for mild to moderate Alzheimer's disease, acting as a selective, reversible competitive acetylcholinesterase inhibitor [44]. The lycorine, haemanthamine, and narciclasine series are leading anticancer bioagents in clinical research [45]. The enormous structural diversity of Amaryllidoideae alkaloids has no equivalent in the Plant Kingdom and can be explained by the chemoecological activity [46].

## 3. Biodiversity and Ecology of Endophytes Associated with Crops Belonging to the Amaryllidaceae Family

The endophytic symbiosis could be an implementation of the microorganisms' strategy aimed at reducing the effects of the external changeable environment through the long-term coevolution with plants providing a stable niche in their tissues [47]. The leaf surface is an attractive habitat for endophytic fungi, which are influenced by the possibility of colonization through the epidermal structures, by leaf health and nutritional status, and by competition with the other microorganisms. Several studies have been carried out to characterize the mycobiota of *A. cepa* rhizosphere and phyllosphere, but much less research has been focused on fungi colonizing internal tissues. Abdel-Gawad et al. [48] isolated and identified, based on macro- and microscopic characters, 24 genera and 38 species of fungi from the rhizoplane of onion, with dominating *Aspergillus* spp., *Cladosporium* spp. and *Penicillium* spp., and 17 genera and 35 species from the phylloplane, with dominating *Aspergillus* and *Penicillium* spp. The root and leaf surface of onion hosted a broader spectrum of species than internal tissues, confirming that plants selectively recruit endophytic microorganisms. Moreover, aboveground plant tissues are exposed to rapid fluctuations in temperature, humidity, and solar radiation, so microorganisms colonizing leaves are also affected by abiotic stress, exceeding sometimes their tolerance thresholds. Abdel-Gawad et al. [48] evidenced that the onion's fungal microbiota dependent on temperature, namely the species *Humicola grisea* (current name *Trichocladium griseum*), *Penicillium mirabile* (current name *Talaromyces verruculosus*), and *Rhizoctonia solani*, were isolated from leaves at 19 °C, whereas other species, such as *Chaetomium brasiliense* (current name *Ovatospora brasiliensis*) and *Zopfiella latipes*, at 28 °C. Moreover, the mentioned species were not specific for onion but isolated from roots of the other crops in the investigated region, namely Assiut Governorate in Egypt. Only one species, *Z. latipes*, was isolated from onion leaves for the first time in Egypt [49]. A red spider lily (*Lycoris radiata*) and golden spider lily (*L. aurea*) are ornamentals of Asian origin, introduced into many countries all over the world because of decorative flowers, but their bulbs are known as poisonous in traditional medicine systems. Zhou et al. [50] identified, using molecular (polymerase chain reaction—PCR) and morphological characteristics, 27 species of fungal endophytes belonging to 14 genera from *L. radiata*. Only *Fusarium* developed hyphae in all organs; *Stagonosporopsis* and *Glomerella* (current name *Colletotrichum*) were isolated from leaf tissues; *Phoma* from the bulb; *Galactomyces*, *Metacordyceps* (current name *Metarhizium*) and *Diaporthe* from root tissues. *Aspergillus*, *Colletotrichum*, *Diaporthe*, *Fusarium*, *Penicillium*, *Phoma*, and *Phyllosticta* were commonly isolated from a wide range of hosts but *Cylindrocarpon*, *Galactomyces*, *Sarocladium*, and *Stagonosporopsis* were described as endophytes of specific plants. In earlier studies, despite the mentioned species, *Trichoderma* sp. was isolated from a bulb of *L. radiata* [51] and *Mucor* sp. from the bulb of *L. aurea* [52]. Notably, *Metarhizium* sp., which was reported as a soil fungus [53], was isolated from *L. radiata* tissues, so this fungus seems to colonize plants occasionally [50].

The relationships between the endophyte fungi and host plant are very diversified and dynamically change from mutualism, symbiosis, and commensalism to pathogenic during plant ontogeny [54,55]. For example, *Colletotrichum*, *Diaporthe*, *Fusarium*, *Phyllosticta*, and *Phoma*, isolated from healthy tissues of *L. radiata* are commonly recognized as pathogenic, so the antifungal alkaloids can enforce symbiotic lifestyle in plants, maintaining a balance between host and its endophytes/parasites. Regarding endophytes colonization during onion's ontogeny, Mueva et al. [56] stated that the seed inoculation was more effective than seedling inoculation in terms of endophytes recovery in subsequent stages of plant development. Indeed, endophytes inoculated at the seed surface colonized seed radicle and plumule and developed internal mycelia in growing tissues. The fungal colonization and distribution

in onion tissues firstly depended on inoculation technique and secondly on the endophyte selection by the host. Independently on the inoculation technique, most of the investigated endophytes, for example *Clonostachys rosea*, *Hypocrea lixii* (current name *Trichoderma lixii*), *Trichoderma asperellum*, *T. atroviride*, *T. harzianum*, and *Fusarium* spp., were isolated from onion roots, followed by stems and leaves. These differences could be due to tissue morphology and physiology, microbiome interactions, and the influence of external conditions [56,57]. Onions have shallow, weakly branched root systems with sparse root hairs, inefficient in the use of soil nutrient resources. The root endophytic and mycorrhizal fungi play a significant role in supporting onions with mineral salts, that is why this species is among the most symbiosis-dependent crops [7]. Focusing on endophytic fungi colonizing shallot roots, Priyadharsini et al. [58] found that the percent of root length with fungal microsclerotia was significantly and negatively correlated with soil phosphorus level. Similarly, percents of root length with dark septate hyphae and dark septate endophyte total colonization were negatively correlated with soil zinc and copper contents. It can be concluded that colonization of shallot roots by fungal endophytes was reduced in soils rich in mineral salts. Wu et al. [59] hypothesized that the endophytic fungal community may be helpful to symbiotic plants (i.a. *Allium mongolicum*) for surviving in the extreme environments of Asian deserts. The mycobiota associated with photosynthesizing or storage leaves, for example *T. harzianum* and *T. koningii*, could act antagonistically to phytopathogens. On the other side, leaves with disease symptoms, with damaged epidermal cells and the lamellar seta shed releasing nutrients, could be secondarily colonized by opportunistic fungi such as *Botrytis cinerea*, *Penicillium aurantiogriseum*, *Alternaria alternata*, and *Cladosporium* spp. [20].

Plant storage tissues, including sugar-rich onions' bulbs, can contain specific endophytes, actively reproducing in these tissues without visible damage [60]. One of the main chemoecological roles of Amaryllidaceae alkaloids is the protection of nutrient-rich bulbs against phytopathogens and herbivores. Xiang et al. [61] isolated and sequenced six fungal endophytes from *Narcissus pseudonarcissus* bulb and only two from leaf tissues. Zhou et al. [19] found that bulbs of *L. radiata* were exclusively colonized at a higher degree than other tissues, probably because of the perennial life cycle of bulbs and annual cycle of other plant parts [62] and because of the space and carbohydrates provided by bulbs as storage sinks [63].

### 3.1. Biochemistry and Functions of Fungal Endophytes Associated with Allioidae Crops

Abdel-Hafez et al. [20] investigated endophytes colonizing *A. cepa* leaves, both healthy and infected by purple blotch (*Alternaria porri*). Fungi were isolated and identified according to their macroscopic and microscopic characteristics. Despite the strains detected from healthy and diseased leaves, belonging to genera *Cladosporium*, *Alternaria*, *Penicillium*, and *Stemphylium*, five species, namely *Absidia corymbifera* (current name *Lichtheimia corymbifera*), *B. cinerea*, *P. aurantiogriseum*, *P. glabrum*, and *Syncephalastrum racemosum*, were isolated only from infected leaves, while three species (*Fusarium oxysporum*, *Trichoderma harzianum*, and *T. koningii*) were isolated only from healthy ones (Table 1). *Trichoderma* spp. showed antagonistic potential against *A. porri*, through competition, lysis, antibiosis, and parasitism [64,65]. The antagonistic effect of *Epicoccum nigrum*, *Penicillium oxalicum*, and *Stachybotrys chartarum* against *A. porri* was antibiosis caused by effective lytic, as well as antimicrobial secondary metabolites produced by endophytic fungi [20]. Previously, Flori and Roberti [66] noticed the antifungal activity of the endophyte *Beauveria bassiana*, inoculated to onion roots, against *F. oxysporum* f. sp. *cepae*, causing basal rot of onion. The antifungal potential of the endophyte *Talaromyces pinophilus* (current name *Penicillium pinophilum*) against *B. cinerea* was described by Abdel-Rahim and Abo-Elyousr [67]. *T. pinophilus* was isolated from onion's inflorescences and identified with PCR amplification of the ribosomal internal transcribed spacer (ITS) region. The mycelium of *T. pinophilus* penetrated intercellularly the hyphae of *B. cinerea*, involving cell wall degrading enzymes (chitinase, lipase, and protease) in the mycoparasitic process.

**Table 1.** Main biological activities of endophytes isolated from Allioidae crops.

| Tissues | Dominating Endophyte Species | Main Activities of the Endophyte | Main Metabolites/Enzymes Linked to Endophyte Bioactivities | Reference |
|---|---|---|---|---|
| *Allium cepa* (leaf) | *Clonostachys rosea, Hypocrea lixii, Trichoderma asperellum, T. atroviride, T. harzianum, Fusarium* sp. | Suppression of *Thrips tabacii* reproduction and viral transmitting | Volatile components | [56,57,68] |
| | *Cladosporium cladosporioides, C. sphaerospermum* | Antifungal against *Alternaria porri* | Not investigated | [20] |
| *A. cepa* (leaf healthy) | *Epicoccum nigrum* | Antifungal against *A. porri* | Flavipin<br> | [20,69] |
| | *Penicillium oxalicum* | Antifungal against *A. porri* | Lytic extracellular enzymes (β-1,3-glucanase, chitinases, cellulases) | [20,70] |
| | *T. harzianum* | Antifungal against *A. porri* | Lytic extracellular enzymes | [20,65] |
| *A. cepa* (leaf infected with *A. porri*) | *Botrytis cinerea, Penicillium aurantiogriseum, Alternaria alternata, Cladosporium* spp. | Antifungal against *A. porri* | Not investigated | [20] |
| *A. cepa* (umbels) | *Talaromyces pinophilus* | Antifungal against *B. cinerea* | Lytic extracellular enzymes (chitinase, lipase, and protease) | [67] |
| *A. cepa* (floral stalks infected with *A. porri*) | *Trihoderma longibrachiatum, T. harzianum,* | Antifungal against *A. porri* | Lytic extracellular enzymes | [64] |
| *A. llium sativum* (leaf) | *Trichoderma brevicompactum* | Antifungal against *Rhizoctonia solani* and *B. cinerea* | 4-acetoxy-12,13-epoxy-9-trichothecene (trichodermin)<br> | [71] |
| *Allium schoenoprasum* (leaves, roots) | *Beauveria bassiana* | Protection of the host | Increased alkaloid level | [72] |

**Table 1.** *Cont.*

| Tissues | Dominating Endophyte Species | Main Activities of the Endophyte | Main Metabolites/Enzymes Linked to Endophyte Bioactivities | Reference |
|---|---|---|---|---|
| *A. schoenoprasum* (bulb) | *Penicillium pinophilum* | Inhibition of the NCI60/ATCC panel of human cancer cell of different tissue origin | Skyrin R=H and Dicatenarin R=OH.  | [73] |
| *Allium filidens* (stem, bulb) | *Aspergillus terreus, Penicillium* sp. | Cytotoxic against carcinoma of the cervix (HeLa), larynx (HEp-2); Inhibition of α-amylase activity | Not investigated | [74,75] |
|  | *A. terreus* | Antibacterial against *Pseudomonas aeruginosa* | Not investigated | [74] |
| *A. filidens* (bud) | *Alternaria tenuissima* | Antibacterial against *P. aeruginosa* and *Staphylococcus aureus* | Not investigated | [74] |
| *Allium longicuspis* (root) | *Aspergillus ochraceus* | Antibacterial against *P. aeruginosa* and *S. aureus* | Not investigated | [74] |
|  | *Aspergillus versicolor* | Antibacterial against *E. coli, P. aeruginosa*, and *S. aureus* | Not investigated | [74] |
|  | *Fusarium* sp. | Antibacterial against *E. coli* | Not investigated | [74] |
| *A. longicuspis* (bulb) | *Aspergillus. spectabilis* | Antibacterial against *P. aeruginosa* and *S. aureus* | Not investigated | [74] |
| *A. longicuspis* (leaf) | *Fusarium sambucinum* | Antibacterial against *E. coli* | Not investigated | [74] |
|  | *Alternaria* sp. | Antibacterial against *E. coli, P. aeruginosa*, and *S. aureus* | Not investigated | [74] |
|  | *A. terreus* | Antibacterial against *E. coli, P. aeruginosa*, and *S. aureus*, inhibition of α-amylase activity | Not investigated | [74,75] |
|  | *Aspergillus flavus* | Antibacterial against *P. aeruginosa* | Not investigated | [74] |

Muvea et al. [56,68] showed the effect of onion inoculation with some strains of endophytic fungi on the proportion of thrips due to reduced feeding and oviposition, caused by antixenotic repellence or higher death rate of thrips. Moreover, the reduced feeding of thrips on endophyte-colonized onions could reduce the transmission of virus diseases, spread by insects.

Among endophytes of garlic that can produce bioactive compounds, Shentu et al. [71] isolated and identified, based on morphological and molecular procedures, *Trichoderma brevicompactum* with strong antifungal activities. Trichodermin, an antifungal compound of *T. brevicompactum* inhibited mycelial growth of *R. solani*, with an $EC_{50}$ of 0.25 µg mL$^{-1}$, and *B. cinerea*, with an $EC_{50}$ of 2.02 µg mL$^{-1}$ (Table 2). A weak inhibition was noted against *Colletotrichum lindemuthianum* ($EC_{50}$ = 25.60 µg mL$^{-1}$). The authors underlined that the relationship between *T. brevicompactum* and the garlic plant remained unclear. Espinoza et al. [72] investigated the chive's growth parameters and secondary metabolites as affected by inoculation with the endophyte fungus *B. bassiana*. The fungus applied to the rhizosphere, colonized plant tissues, and finally was isolated from roots and leaves, affecting total alkaloids content but not leaves yield. Koul et al. [73] isolated and morphologically and molecularly identified the fungus *Penicillium pinophilum*, from bulbs of chive's population native to snow mountain regions of India. *P. pinophilum* was a source of anticancer anthraquinones, dicatenarin, and skyrin. Both compounds inhibited human pancreatic cancer (MIA PaCa-2) cells with least $IC_{50}$ values of 12 µg mL$^{-1}$ and 27 µg mL$^{-1}$ respectively, through mitochondrial-mediated apoptotic pathway. Dicatenarin cytotoxic/proapoptotic activity was more pronounced than skyrin due to the presence of an additional phenolic hydroxyl group at C-4, which increased reactive oxygen species generation [73].

Wild and endemic *Allium* species were also the object of investigation. In the latter respect, Abdulmyanova et al. [74] screened *Allium filidens* Regel and leaves of *A. longicuspis* Regel regarding endophytic fungi biodiversity and bioactivity. Among 16 isolates of endophytic fungi obtained from these plants and identified morphologically, the highest biodiversity was determined for bulbs of *A. filidens* and leaves of *A. longicuspis*. The *Penicillium* spp. were the most dominant symbionts of *A. filidens*, while *Aspergillus* spp. were commonly isolated from *A. longicuspis*. Beside cosmopolitan species, the rare endophytes *Alternaria tenuissima*, *Aspergillus spectabilis*, and *Cladosporium tenussimum* were also isolated. The endophytic fungi detected in the same host varied regarding bioactivity. For example, three strains of *Penicillium* sp. isolated from bulbs of *A. filidens* were different in cytotoxic, antibacterial, and antiamylase activity, two strains of *Alternaria* sp. from leaves of *A. longicuspis* exhibited only antibacterial activity [74,75]. Bulbs of both described *Allium* species, endemic in Afganistan, have been used in traditional Asian medicine [76].

### 3.2. Biochemistry and Functions of Fungal Endophytes Associated with Amaryllidoideae Crops

Amarylidoideae alkaloids can be involved in chemical crosstalk between host plant and endophytes as communication molecules that are responsible for the shaping of plant-microbe interactions. This phenomenon was more widely investigated for endophytic bacteria, which can promote the synthesis of Amaryllidaceae alkaloids [77,78], but endophytic fungi are also involved in plant-endophyte and endophyte–endophyte interspecies communication. For example, Wang et al. [79] investigated endophytic fungi and bacteria in the bulbs of the Chinese sacred lily (*Narcissus tazetta*), widely used as an ornamental and medicinal plant in Asia [79]. The authors defined nine hexacyclopeptides produced by fungus and selectively accumulated by an endophytic bacterium *Achromobacter xylosoxidans* isolated from the same tissue (Table 2). The production of targeted hexacyclopeptides by *F. solani* was possible only in planta and decreased in vitro conditions. However, the ecological basis of this chemical cross-talk needs future investigations. Yang et al. [80] isolated and identified, using morphological and molecular methods, 18 strains of endophytic fungi from *Narcissus* sp. Three species, particularly *Rhinocladiella* sp., demonstrated significant inhibitory activity against acetylcholinesterase.

Onofri et al. [81] identified, using conventional taxonomic techniques, four strains of *Cryptococcus laurentii* (current name *Papiliotrema laurentii*), C1–C3 from root tips, C4 from outer bracts of bulb of daffodil (*N. pseudonarcissus*). The authors observed that lycorine, an alkaloid of *Narcissus* bulbs,

inhibited the growth of C1–C3 but not C4 strains of fungi. The inhibition was due to destroying the cellular membranes and interfering with the substrate absorption and cell metabolism, namely blocking L-galactonic acid $\gamma$-lactone conversion into ascorbate by lycorine. In contrast, *C. laurentii*, isolated from the lycorine-containing bracts of the bulb, was able to degrade lycorine and to use decomposition products as growth stimulators.

*L. radiata* is the main source of Amaryllidaceae alkaloids but the low yield and high costs, resulting from its complex procedures and mixed stereoisomers, limit pharmaceutical development of plant-delivered drugs [50]. *Lycoris* spp. were objects of some investigations regarding endophyte microbiota and assessment of the biological activity of their metabolites. *Penicillium* sp. isolated from *L. aurea* was able to produce galanthamine in vitro [82], and the other nonidentified fungus strain L-10 possessed antibacterial and antifungal activity against *Staphylococcus aureus* and *Candida albicans*, respectively [52]. This phenomenon confirmed the antagonism between fungal and bacterial symbionts of this plant. Both novel and known compounds, especially alkaloids, could be produced by in vitro grown endophytic fungi isolated from *Lycorsis* spp. bulbs. Moreover, the inoculation with fungal endophytes enhanced the level of various alkaloids in *L. radiata*. So, inoculation with particular fungus or consortium of fungi can be used for increasing the content of targeted alkaloids during plant cultivation [50].

Li et al. [83] investigated drimane-type sesquiterpenoids of *Aspergillus versicolor*. Among the latter compounds, Versicalin A showed moderate cytotoxic activity against HL-60 tumor cells with an $IC_{50}$ value of 5.6 μM, while proversilin C and E showed moderate cytotoxicity against human tumor HL-60, SMMC-7721, A-549, MCF-7, and SW-480 cell lines and the normal colonic epithelial cells NCM460 with $IC_{50}$ values ranging from 7.3 to 28.4 μM [83,84]. The synthesis of the same chemical compounds by the plant host and endophytic fungus is the phenomenon described for some other species as an example of highly specified coevolution. Moreover, this phenomenon has a great significance in the detection and production of pharmaceutically valuable plant/endophyte derived drugs [85–87].

**Table 2.** Main biological activities of endophytes isolated from Amaryllioidae crops.

| Tissues | Dominating Endophyte Species | Main Activities of the Endophyte | Main Metabolites/Enzymes Linked to Endophyte Bioactivities | Reference |
|---|---|---|---|---|
| *Narcissus pseudonarcissus* (root tips, outer bracts) | *Papiliotrema laurentii* | Antifungal against some strains of *Cryptococcus laurentii* | Lycorine  | [81] |
| *Narcissus* sp. | *Rhinocladiella* sp. | Inhibition of acetylcholinesterase | Not investigated | [80] |
| *Narcissus tazetta* (bulb) | *Fusarium solani* | Selective accumulation by an endophytic bacterium, *Achromobacter xylosoxidans* | Hexacyclopeptides | [79] |
| *L. aurea* | *Penicillium* sp. | Not investigated | Galantamine  | [82] |
| *L. radiata* (bulb) | *Aspergillus versicolor* | Cytotoxic against human tumor HL-60 cells with an IC50 value of 5.6 lM.15 | (±)–Versicalin A, keto-enol tautomer (+)–6a  | [83] |

**Table 2.** *Cont.*

| Tissues | Dominating Endophyte Species | Main Activities of the Endophyte | Main Metabolites/Enzymes Linked to Endophyte Bioactivities | Reference |
|---|---|---|---|---|
| *L. radiata* (bulb) | *A. versicolor* | Cytotoxic against human tumor HL-60, SMMC-7721, A-549, MCF-7, and SW-480 cell lines | Proversilin C  Proversilin E  | [84] |
| *Eucharis cyaneosperma* | *Streptomyces recifensis* | Not investigated | Not investigated | [85] |

## 4. Conclusions

The increasing demand for Amaryllidaceae crops is triggered not only by traditional culinary usage but also by cultural attitudes, social beliefs, modern interest in exotic and ethnic foods, and medicinal applications. However, secondary metabolites responsible for wide applications of crops in every area of human activity are synthesized by both plant and its endopytic microbiota. Indeed, endophytic fungi associated with this group of plants provide host plants with nutrients and water, alleviate biotic and abiotic stresses, increase stress tolerance, and affect metabolome profile. They are a source of metabolites of antifungal and antiparasitic activity and have a promising perspective in the application as effective biocontrol agents, replacing chemical fungicides and pesticides. Moreover, as the chemosynthesis of Amaryllidaceae alkaloids needs complicated and costly procedures, plants remain an exclusive source of these alkaloids for the pharmaceutical industry. Symbiotic endophytic fungi can be used to increase alkaloids yield in plants or as an alternative source of alkaloids and other bioactive compounds in vitro cultures. Taking into account the scant research on endophytic fungi associated with Amaryllidaceae as a prolific source of phytochemicals, the need has raised for screening investigations aimed to identify the endophytic species, as well as the molecular and genetic basis of their relationship with the host plants.

## 5. Review Methodology

The present review was based on the literature collected from the leading life science databases, including AGRICOLA, AGRIS, BioOne, CAB Abstracts, PubMed, SciELO, Scopus, and Web of Science. Bibliometric analysis was used for the review, evaluation, and objective representation of the structure within a presented research area, namely Amarylidaceae–fungal endophytes relationship. The most relevant aspects of the evolution, advances, and trends in the reviewed field were presented [88]. References were collected, studied, and selected considering (i) the reports of endophytes isolated from the species of Amaryllidaceae family, (ii) the reports of therapeutic utilization of the host plant or/and an endophyte, (iii) the effects of host plant phylogeny on root microbiome assembly. Plant names were verified according to the Global Biodiversity Information Facility [89] and The Plant List [90]; endophyte taxa were verified according to MycoBank database [91].

**Author Contributions:** Conceptualization, A.S., and G.C.; writing—original draft preparation, A.S., and G.C.; writing—review and editing, N.G., M.T.A., A.T.; visualization, A.S. and A.T.; supervision, A.S., and G.C. All authors have read and agreed to the published version of the manuscript.

**Funding:** This study was partially supported by the Ministry of Science and Higher Education of the Republic of Poland.

**Conflicts of Interest:** The authors declare no conflict of interest.

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
