# Peer review of "Biodiversity, Ecology, and Secondary Metabolites Production of Endophytic Fungi Associated with Amaryllidaceae Crops"

_agriculture, doi:10.3390/agriculture10110533_

Round 1

Reviewer 1 Report

The review presented here addresses plant-associated endophytic fungi, their importance in agriculture and the uses of these species as a potential source of active compounds.

Microbiome research is becoming a hot topic in the field of microbiology, and reviews that summarize the state of the art on specific fields are certainly welcome. In this particular case, the authors focused on endophytic fungi associated with the Amaryllidaceae family.

Overall, the review is well organized, but hard enough to read. English is grammatically good but personally I don't like the written style. In many cases the authors do not use adverbs that could help the reading flow. An example is given between lines 123-129, where the authors list some organic sulfur compounds and their properties, but as separate sentences. This sounds very "mechanical" to me.

Beyond that, there are only a few minor issues that should be fixed:

Table 1: I suggest to draw dicatenarin and skyrin structure differently: you may add the structure once reporting R=H skyrin and R=OH dicatenarin.

Line 302: …”antibacterial activity against Staphylococcus aureus and Candida albicans”. Better “antibacterial and antifungal activity against Staphylococcus aureus and Candida albicans respectively”

Lines 320-324: I could not understand the connection between “The increasing demand for Amaryllidaceae crops” and “the endophytic fungi associated with this group of plants”.

I would invert the order of paragraph 4 and 5.

Author Response

Reviewer 1:

The review presented here addresses plant-associated endophytic fungi, their importance in agriculture and the uses of these species as a potential source of active compounds.

Microbiome research is becoming a hot topic in the field of microbiology, and reviews that summarize the state of the art on specific fields are certainly welcome. In this particular case, the authors focused on endophytic fungi associated with the Amaryllidaceae family.

Overall, the review is well organized, but hard enough to read. English is grammatically good but personally I don't like the written style. In many cases the authors do not use adverbs that could help the reading flow. An example is given between lines 123-129, where the authors list some organic sulfur compounds and their properties, but as separate sentences. This sounds very "mechanical" to me.

Authors response: the English style and grammar have been improved.

Beyond that, there are only a few minor issues that should be fixed:

Table 1: I suggest to draw dicatenarin and skyrin structure differently: you may add the structure once reporting R=H skyrin and R=OH dicatenarin.

Authors response: corrected.

Line 302: …”antibacterial activity against Staphylococcus aureus and Candida albicans”. Better “antibacterial and antifungal activity against Staphylococcus aureus and Candida albicans respectively”

Authors response: addressed.

Lines 320-324: I could not understand the connection between “The increasing demand for Amaryllidaceae crops” and “the endophytic fungi associated with this group of plants”.

Authors response: we included the sentence: “However, secondary metabolites promoting wide applications of Amaryllidaceae crops in every area of human activity, are synthesized by both plants and their endopytic microbiota.”

I would invert the order of paragraph 4 and 5.

Authors response: addressed.

We would like to thank the Reviewer for the thoughtful comments and efforts aimed to improve our manuscript.

Reviewer 2 Report

The authors reviewed available information on studies of endophytic fungi in Amaryllidaceae species. 

Their idea was to highlight the biodiversity/chemical potential that endophytic fungi identified in these species had and how they could be used by the pharmaceutical industries. Since the family is known to have ornamental and food-related crops, it makes sense for them to consider a more in-depth analysis of a family with the potential for biotechnology at a lower cost. 

The Amaryllidaceae are not as recognized as they should, and I think it would be a stronger proposal of a review if the objective would have been other than just listing what is known so far without a "sharper" purpose of use. Still, it is very informative and necessary as a readily available compilation of information and proposal for future model studies that may have scape other's eyes. 

Although the focus seems to be put mostly on onion and chives (I guess are the species with the most information), I think a projection of what other species (that are later mentioned almost at the end of the paper) could do and how they interact with their microbiome would have strengthened the work presented. 

Since it was mentioned that identifying fungal endophytes was important to truly be applied in agriculture to help regulate or stimulate the production of certain secondary metabolites, I think highlighting more what is known about the microbiome of some of these plants would have been very helpful.

Explaining how the composition in each organ differs and how the presence of some endophytes is stronger in the underground organs of the plants is really important, and I think a better explanation of how some of these fungi could modulate changes in the rest of the organs of the plant would have been interesting.

The methods used did not highlight if the software was used for identifying some of the main/keywords searches. Nowadays, there are great bibliometric tools that could be used and that would improve the quality of the paper.

Fungal endophytes are of extreme interest due to their biochemical compounds, so I think it was a great decision to focus on them, still, it was disconnected from understanding how it would interact with the rest of the microbiome of these plants. 

Monocots are more complex plants to study but of extreme potential. I applaud their highlight of the recovery of some of these endophytes, but I would have like to understand better (in the text) if the identification was done using sequencing or was just by isolation (since this would limit future applications).

Overall, I believe that the idea and objective are important and we should start looking at species that are available, affordable, and that has more than one purpose. I agree with the selection of the Family to be reviewed.

Author Response

Reviewer 2:

The authors reviewed available information on studies of endophytic fungi in Amaryllidaceae species. 

Their idea was to highlight the biodiversity/chemical potential that endophytic fungi identified in these species had and how they could be used by the pharmaceutical industries. Since the family is known to have ornamental and food-related crops, it makes sense for them to consider a more in-depth analysis of a family with the potential for biotechnology at a lower cost. 

The Amaryllidaceae are not as recognized as they should, and I think it would be a stronger proposal of a review if the objective would have been other than just listing what is known so far without a "sharper" purpose of use. Still, it is very informative and necessary as a readily available compilation of information and proposal for future model studies that may have scape other's eyes. 

Although the focus seems to be put mostly on onion and chives (I guess are the species with the most information), I think a projection of what other species (that are later mentioned almost at the end of the paper) could do and how they interact with their microbiome would have strengthened the work presented. 

Authors response: Thank you for the positive opinion of the manuscript, and of the idea we developed. As you mentioned, only some species of Amaryllidaceae were described in connection with their endophyte microbiota. We focused on well documented sources to avoid speculation, but based on your recommendations we have complemented the text with more detailed information regarding the endophyte-host plant interactions.

Since it was mentioned that identifying fungal endophytes was important to truly be applied in agriculture to help regulate or stimulate the production of certain secondary metabolites, I think highlighting more what is known about the microbiome of some of these plants would have been very helpful.

Authors response: in addition to endophytes, we mentioned the most important aspect of plant-microbiota relations, namely micorrhiza [see References 5-8], which was reviewed previously for some Amarylidaceae genera/species. We addressed this review to the Special Issue of Agriculture dedicated to fungal endophytes, so we highlighted all the available reports focusing on this topic to our references.

Explaining how the composition in each organ differs and how the presence of some endophytes is stronger in the underground organs of the plants is really important, and I think a better explanation of how some of these fungi could modulate changes in the rest of the organs of the plant would have been interesting.

Authors response: We complemented the text with more detailed information regarding the interspecies relations among microbiota colonizing different plant parts. Based on the available literature, we have narrow knowledge of fungal endophytes colonizing aboveground parts of Amarylidaceae plants. In the latter respect, the reason is connected to the biology of this family crops, namely the short life of particular green leaves and the susceptibility to fungal diseases which represent a major  field of investigations because of the economic importance. We tried to reflect the state of knowledge concerning the  endophyte tissue-dependent colonisation of the Amaryllidaceae plants, which is mostly related to soil microbiome, and that is why the endophytes isolated from roots and bulbs were described more widely.

The methods used did not highlight if the software was used for identifying some of the main/keywords searches. Nowadays, there are great bibliometric tools that could be used and that would improve the quality of the paper.

Authors response: We supplemented the Material and Methods section with the method of bibliometric analysis with the most adequate reference. In order to provide a broader explanation, we assessed the scientific literature concerning the field of review through a bibliometric analysis (10.1007/s10489-017-1105-y; 10.1080/09737766.2012.10700939,  10.1177/1094428114562629). Because of not numerous research focused on the analysed area (a few research teams working with a few species in the case of present review; or investigations on species of different families from particular regions), the data visualization did not show  results which could have significantly contributed to this review.

Fungal endophytes are of extreme interest due to their biochemical compounds, so I think it was a great decision to focus on them, still, it was disconnected from understanding how it would interact with the rest of the microbiome of these plants. 

Authors response: We complemented the text with more detailed information regarding the interspecies relations within microbiota. Additionally, their interactions (antibacterial, antifungal activity) have been presented in Tables together with endophytes linking to particular plant parts.

Monocots are more complex plants to study but of extreme potential. I applaud their highlight of the recovery of some of these endophytes, but I would have like to understand better (in the text) if the identification was done using sequencing or was just by isolation (since this would limit future applications).

Authors response: We have now provided the method of fungi identification.

 Overall, I believe that the idea and objective are important and we should start looking at species that are available, affordable, and that has more than one purpose. I agree with the selection of the Family to be reviewed.

Authors response: Thank you for the professional review and helpful comments.